# Association between confirmed congenital Zika infection at birth and outcomes up to 3 years of life

Najeh Hcini[1,2], Yaovi Kugbe[3], Zo Hasina Linah Rafalimanana[3], Véronique Lambert[1], Meredith Mathieu[1], Gabriel Carles[1], David Baud [4], Alice Panchaud[5,6,7] & Léo Pomar [1,4,7 ✉]

Little is known about the long-term neurological development of children diagnosed with congenital Zika infection at birth. Here, we report the imaging and clinical outcomes up to three years of life of a cohort of 129 children exposed to Zika virus in utero. Eighteen of them (14%) had a laboratory confirmed congenital Zika infection at birth. Infected neonates have a higher risk of adverse neonatal and early infantile outcomes (death, structural brain anomalies or neurologic symptoms) than those who tested negative: 8/18 (44%) vs 4/111 (4%), aRR 10.1 [3.5–29.0]. Neurological impairment, neurosensory alterations or delays in motor acquisition are more common in infants with a congenital Zika infection at birth: 6/15 (40%) vs 5/96 (5%), aRR 6.7 [2.2–20.0]. Finally, infected children also have an increased risk of subspecialty referral for suspected neurodevelopmental delay by three years of life: 7/11 (64%) vs 7/51 (14%), aRR 4.4 [1.9–10.1]. Infected infants without structural brain anomalies also appear to have an increased risk, although to a lesser extent, of neurological abnormalities. It seems paramount to offer systematic testing for congenital ZIKV infection in cases of in utero exposure and adapt counseling based on these results.

---

[1] Department of Obstetrics and Gynaecology, West French Guiana Hospital Center, French Guiana, France. [2] CIC Inserm 1424, Department of Health Training and Research, University of French Guiana, French Guiana, France. [3] Department of Pediatrics, West French Guiana Hospital Center, French Guiana, France. [4] Materno-Fetal and Obstetrics Research Unit, Department "Woman-Mother-Child", Lausanne University Hospital and University of Lausanne, Lausanne, Switzerland. [5] Service of Pharmacy, Lausanne University Hospital and University of Lausanne, Lausanne, Switzerland. [6] Institute of Primary Health Care (BIHAM), University of Bern, Bern, Switzerland. [7] These authors contributed equally: Alice Panchaud, Léo Pomar. ✉email: leo.pomar@chuv.ch

In the last decade, ZIKV has spread through the Pacific Islands[1] and the Americas[2], leading to a worldwide epidemic. It is now well demonstrated that ZIKV is associated with multiple congenital abnormalities, particularly affecting the central nervous system[3–5]. Long-term disabilities, including cerebral palsy, epilepsy, and neurosensory alterations have been described in infants from the American and Brazilian cohorts[6–9]. Infants included in these cohorts were mostly affected by congenital Zika Syndrome (CZS) with cerebral anomalies or neurological impairments at birth. The evolution of those with a laboratory-confirmed congenital infection but who were asymptomatic and without cerebral anomalies at birth remains poorly described. Overall, the correlation between laboratory testing for congenital ZIKV infection at birth and long-term disability including sensory and cognitive deficits is still lacking and may have major clinical and public health implications[10].

The French Guiana Western Hospital Center (CHOG, Centre Hospitalier "Franck Joly", referral center for western French Guiana), was confronted with the ZIKV epidemic from January to September 2016. In a previous study of a cohort of ZIKV-infected pregnant women followed at the CHOG, we estimated that 279 infants were liveborn to these mothers[11]. Here, using a subset of these infants, we show that those with a laboratory-confirmed congenital Zika infection at birth have higher risks of imaging and clinical adverse outcomes at 2 months, 2 years, and 3 years of life than those tested negative at birth.

## Results

**Study population.** Between January and September 2016, 132 newborns from 128 ZIKV-infected mothers (four dichorionic twins) were enrolled for prospective follow-up at the pediatric clinic of the CHOG. Three of them were not tested for ZIKV before postpartum discharge and were excluded. Among 129 children tested for ZIKV at birth, 18 (14.0%) had a laboratory-confirmed congenital infection and 111 (86.0%) tested negative (details of fetal and neonatal testing are presented in Supplementary Table 1, Fig. 1 describes the enrollment).

**Baseline characteristics.** Median maternal age at delivery was 25 and 26 years old in the group of congenital infections and the negative group, respectively. Maternal infection diagnosed in the first trimester of pregnancy was more frequent in mothers of infected newborns (38.9% vs 24.3%). Cesarean section delivery occurred more often in these mothers (27.8% vs 12.6%). Twins were also more frequent among infected newborns (11.1% vs 5.4%). Alcohol and drug consumption was more frequent in mothers of newborns that tested negative for ZIKV at birth (10.8% vs 5.6%). Other environmental or recreative exposures and co-morbidities during pregnancy including low socioeconomic status, infant sex, rate of prematurity, and neonatal adaptation were not different between these groups (Table 1). Three mothers had an identified TORCH (toxoplasmosis, rubella, cytomegalovirus, and Herpes simplex virus) co-infection during pregnancy, however, newborns tested negative during the pregnancy and at birth (one primary toxoplasmosis and two primary cytomegalovirus infections).

**Neonatal and early infantile outcomes.** Among 18 neonates with a confirmed congenital ZIKV infection at birth, 8 (44.4%) were identified to have an adverse outcome during their first 2 months of life: 6 (33.3%) had severe structural brain anomalies, of whom one died in his first day of life (CZS with arthrogryposis and severe brainstem dysfunction) and two others had severe neurological symptoms. The risk of adverse outcomes at 2 months was higher for infected infants compared with those tested

### Table 1 Baseline characteristics of ZIKV-exposed pregnancies and newborns.

| Characteristics | Confirmed congenital infection at birth | Negative testing at birth | Std diff |
|---|---|---|---|
| | **N = 18** | **N = 111** | |
| Maternal age at birth (years)—median (min–max) | 25 (18–38) | 26 (18–43) | 0.09 |
| *Maternal socioeconomic status—no. (%)* | | | |
| Low | 6 (33.3%) | 38 (34.2%) | −0.02 |
| Moderate | 8 (44.4%) | 46 (41.4%) | 0.06 |
| High | 1 (5.6%) | 8 (7.2%) | −0.07 |
| Unknown | 3 (16.7%) | 19 (17.1%) | −0.01 |
| *Maternal exposure during pregnancy—no. (%)* | | | |
| Alcohol consumption | 1 (5.6%) | 12 (10.8%) | −0.19 |
| Drug use | 0 (0.0%) | 1 (0.9%) | 0.13 |
| Current smoker | 1 (5.6%) | 5 (4.5%) | 0.05 |
| Lead poisoning | 1 (5.6%) | 6 (5.4%) | 0.01 |
| *Any maternal co-morbidities[π]—no (%)* | 3 (16.7%) | 15 (13.5%) | 0.09 |
| Diabetes (previous or gestational) | 1 (5.6%) | 7 (6.3%) | −0.03 |
| Vascular pathologies | 1 (5.6%) | 5 (4.5%) | 0.05 |
| Severe anemia | 1 (5.6%) | 5 (4.5%) | 0.05 |
| Co-infections* | 1 (5.6%) | 2 (1.8%) | 0.20 |
| *Maternal Zika infection—no. (%)* | | | |
| Symptomatic | 4 (22.2%) | 22 (19.8%) | 0.06 |
| Asymptomatic | 14 (77.8%) | 89 (80.2%) | −0.06 |
| *Trimester of maternal Zika infection diagnosis—no. (%)* | | | |
| T1 infection | 7 (38.9%) | 27 (24.3%) | 0.31 |
| T2 infection | 7 (38.9%) | 37 (33.3%) | 0.11 |
| T3 infection | 2 (11.1%) | 25 (22.5%) | −0.30 |
| Unknown | 2 (11.1%) | 22 (19.8%) | −0.24 |
| Dichorionic twins | 2 (11.1%) | 6 (5.4%) | 0.20 |
| *Fetus gender—no. (%)* | | | |
| Female | 10 (55.6%) | 63 (56.8%) | −0.02 |
| Male | 8 (44.4%) | 48 (43.2%) | 0.02 |
| *Term at birth (weeks' gestation)—median (min–max)* | 39 (32–41) | 39 (36–41) | −0.18 |
| Premature birth <37 wg | 1 (5.6%) | 6 (5.4%) | 0.01 |
| *Mode of delivery—no. (%)* | | | |
| Normal | 13 (72.2%) | 97 (87.4%) | −0.38 |
| C-section | 5 (27.8%) | 14 (12.6%) | 0.38 |
| *Neonatal adaptation* | | | |
| Abnormal Apgar score (<6 at 5 min)—no (%) | 2 (11.1%) | 13 (11.7%) | −0.02 |
| Abnormal lactate level (> 4.5)—no (%) | 2 (11.1%) | 16 (14.4%) | −0.09 |

Age, weight, and term variables are presented as a median with extreme values. For qualitative variables, absolute frequencies and relative frequencies (percentages) are presented. Baseline characteristics are compared between groups using standardized differences.
Alcohol consumption was defined as ongoing consumption during the pregnancy after its diagnosis. Current smoking was defined as ongoing smoking during pregnancy after its diagnosis. Baseline characteristics were considered unbalanced if standardized differences (Std diff) were > 0.1.
[π]including multiple maternal co-morbidities.
*One primary cytomegalovirus infection in the infected group; one primary CMV and one primary toxoplasmosis infection in the control group.

negative at birth (4/111, 3.6%), even after adjustment for maternal infection diagnosed in the first trimester of pregnancy: aRR 10.1 [95% CI 3.5–29.0]. When considering only infants without structural brain anomalies, the risk of severe neurological symptoms was also higher in those diagnosed with a congenital ZIKV infection (2/12, 16.7% vs 1/108, 0.9%): aRR 20.5 [95% CI 2.0–207.7] (Table 2 and Fig. 2). At 2 months of life, microcephaly,

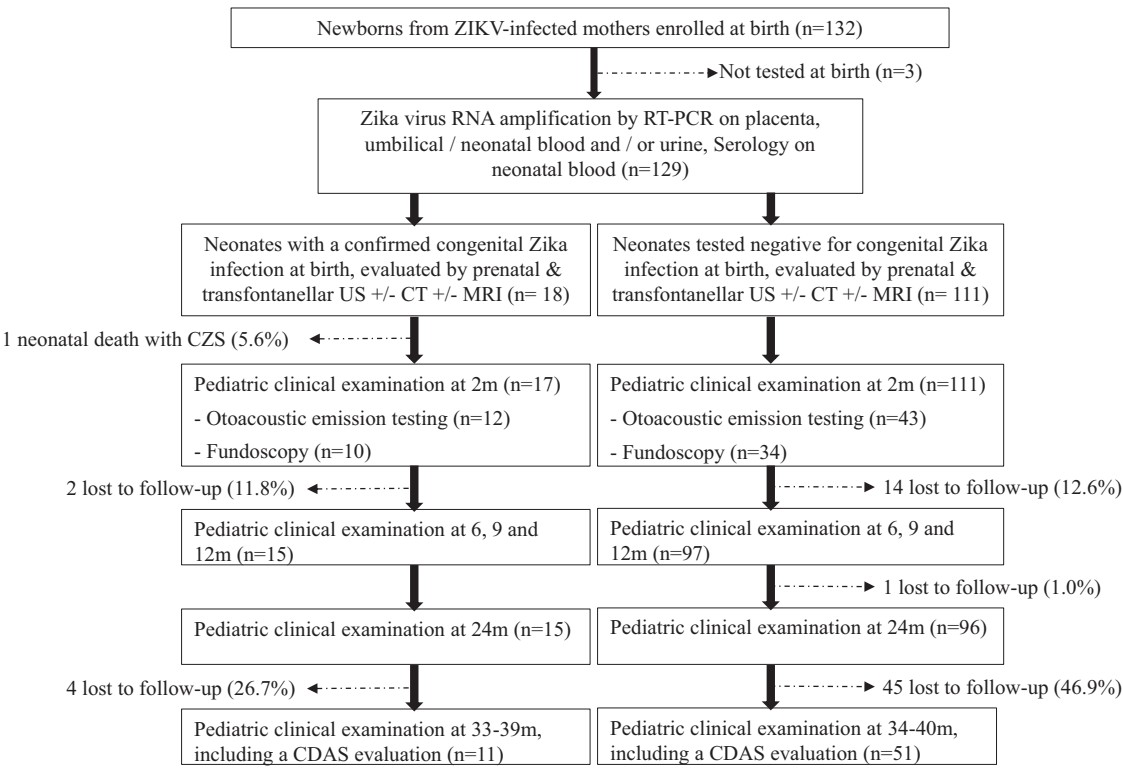

**Fig. 1 Flowchart of newborns exposed to Zika virus in utero.** All newborns from Zika-infected mothers, living in western French Guiana and followed at the pediatric clinic of the CHOG, were enrolled in this cohort following an informed consent process. At birth, they underwent clinical examination (including anthropometric measurements and a special focus on the neurological status), transfontanellar ultrasound (US), and testing for congenital Zika infection (PCR on urine, blood, and placenta; serology; and testing in cerebrospinal fluid if symptomatic). After postpartum discharge, they were recalled at 2, 6, 9, 12, 18, and 24 months of life for a pediatric examination. At 3 years of life (33–42 months), they were recalled for an evaluation of their development using the Child Development Assessment Scale (CDAS).

abnormal evoked otoacoustic emission (OAE) testing, and ocular anomalies were confirmed in 2/17 (11.8%), 2/12 (16.7%), and 3/10 (30%) infected infants; and in 1/111 (0.9%), 1/43 (2.3%), and 2/34 (5.9%) infants that tested negative at birth, respectively (Table 3). Head circumference (HC) and weights, according to the results of congenital Zika infection testing, are presented in Fig. 3.

**Outcomes at 2 years of life.** At 2 years of life, 15 children with a confirmed congenital ZIKV infection at birth and 96 who tested negative at birth were still followed at the CHOG pediatric clinic.

Among infected children, 5/15 (33.3%) had neurologic impairments: 2 with cerebral palsy, 3 with severe dystonia, and 3 with seizures. Two of those with neurologic impairments had motor acquisition delays, partial, or complete blindness, and one had hearing deficits. Hearing impairment was also diagnosed in another infected child, without neurologic impairments (Table 4). Overall, the risk of adverse outcomes at 2 years of life was higher in infected children (6/15, 40.0%) compared with those that tested negative at birth (5/96, 5.2%), even when only considering children without structural brain anomalies (2/10, 20.0% vs 3/93, 3.2%): aRR 6.7 [95% CI 2.2–20.0] and aRR 6.2 [1.2–33.0], respectively (Table 2 and Fig. 2).

**Neurodevelopment at 3 years of life.** Eleven (11/17, 64.7%) children of the infected group and 51 (51/111, 45.9%) children of the group that tested negative at birth presented for neurodevelopmental screening in August and September 2019. The median age at evaluation was 35 months in the infected group and 36 months in the group of children that tested negative at birth.

A developmental score below −2SD ("Referral" zone) in at least one domain was observed in 7/11 (63.6%) infected children, the cognitive and language domain being the most affected (6/11, 54.5%). Details of the Child Development Assessment Scale (CDAS) are presented in Table 4 and Fig. 4. Children with a confirmed congenital ZIKV infection at birth had a higher risk of a CDAS score in the "referral zone" (<−2SD) compared with children that tested negative (7/51, 13.7%), even when considering only those without structural brain anomalies: aRR 4.4 [1.9–10.1] and aRR 3.0 [1.0–9.0], respectively (Table 2 and Fig. 2).

Children with a CDAS score in the "referral zone" were the same who had an adverse outcome at 2 years among the infected. In addition, one infected infant without structural brain anomalies and who remained asymptomatic until 2 years of age was screened as at risk for developmental delay at 3 years of life.

**Effect-modifiers.** Exposures during pregnancy, maternal age, co-morbidities, socioeconomic status, infant sex, twins, prematurity, and the mode of delivery were tested as effect-modifiers on main outcomes, and no interactions were identified. The presence of structural brain anomalies, however, was an effect-modifier for severe neurological symptoms at 2 months of life, and on adverse outcomes at 2 and 3 years of life. A sub-analysis of children with and without structural brain anomalies is presented in Table 2.

## Discussion
**Main results.** In this study, we assessed the development of children with laboratory-confirmed congenital ZIKV infection up to 3 years of age. Our results indicate that infected neonates have a higher risk of neurological symptoms at birth (27.8%),

**Table 2 Main outcomes and associations with congenital ZIKV infection at birth.**

| Main outcomes | Confirmed congenital infections | Negative neonatal testing | RR [95% CI] | p | aRR* [95% CI] | p |
|---|---|---|---|---|---|---|
| *Neonatal and early infantile adverse outcomes*^a | 8/18 (44.4%) | 4/111 (3.6%) | 12.3 [4.1–36.8] | <0.001 | 10.1 [3.5–29.0] | <0.001 |
| Neonatal demise | 1/18 (5.6%) | 0/111 (0.0%) | | 0.14 | | |
| Structural brain anomalies | 6/18 (33.3%) | 3/111 (2.7%) | 12.3 [3.4–45.0] | <0.001 | 10.5 [3.0–36.7] | <0.001 |
| Severe neurologic symptoms | 5/18 (27.8%) | 1/100 (0.9%) | 30.8 [3.8–248.9] | 0.001 | 26 [3.2–208.3] | 0.002 |
| In infants with structural brain anomalies | 3/6 (50.0%) | 0/3 (0.0%) | | 0.238 | | |
| In infants without structural brain anomalies | 2/12 (16.7%) | 1/108 (0.9%) | 18 [1.8–184.1] | 0.015 | 20.5 [2.3–184.3] | 0.007 |
| *Adverse outcomes at 2 years of life*^b | 6/15 (40.0%) | 5/96 (5.2%) | 7.7 [2.7–22.1] | <0.001 | 6.7 [2.2–20.0] | 0.001 |
| In children with structural brain anomalies | 4/5 (80.0%) | 2/3 (66.7%) | 1.2 [0.48–3.0] | 0.695 | 1.1 [0.4–2.7] | 0.845 |
| In children without structural brain anomalies | 2/10 (20.0%) | 3/93 (3.2%) | 6.2 [1.2–32.8] | 0.032 | 6.2 [1.2–33.0] | 0.032 |
| Neurologic impairments | 5/15 (53.3%) | 4/96 (4.2%) | 8.0 [2.4–26.5] | 0.001 | 7.1 [2.1–24.1] | 0.002 |
| Delay in motor acquisitions | 2/15 (13.3%) | 1/96 (1.0%) | 12.8 [1.2–132.6] | 0.033 | 9.0 [0.81–99.6] | 0.073 |
| Neurosensory alterations | 3/15 (20.0%) | 1/96 (1.0%) | 19.4 [2.1–172.7] | 0.008 | 22.5 [2.5–205.5] | 0.006 |
| *Referral for suspicion of neurodevelopment < −2SD in at least one domain at 3 years of life*^c | 7/11 (63.6%) | 7/51 (13.7%) | 4.6 [2.0–10.5] | <0.001 | 4.4 [1.9–10.1] | <0.001 |
| In children with structural brain anomalies | 4/4 (100.0%) | 0/2 (0.0%) | | 0.067 | | |
| In children without structural brain anomalies | 3/7 (42.9%) | 7/49 (14.3%) | 3.0 [1.0–9.0] | 0.05 | 3.0 [1.0–9.0] | 0.049 |
| Motor | 2/11 (18.2%) | 1/51 (2.0%) | 9.3 [0.9–93.4] | 0.059 | 9.3 [1.0–88.4] | 0.051 |
| Cognitive and language | 6/11 (54.5%) | 3/51 (5.9%) | 9.3 [2.7–31.5] | <0.001 | 8.3 [2.4–28.2] | 0.001 |
| Socio-affective | 4/11 (36.4%) | 6/51 (11.8%) | 3.1 [1.0–9.1] | 0.041 | 3.3 [1.2–9.0] | 0.021 |

^a 129 infants evaluated from birth to 2 months of life.
^b 111 children evaluated up to 2 years of life.
^c 62 children evaluated at 3 years of life using the Child Development Assessment Scale.
*Adjusted on the trimester of maternal Zika infection. Exposures during pregnancy, maternal age, socioeconomic status and co-morbidities, infant gender, twins, prematurity, and the mode of delivery were tested as effect-modifiers for severe neurological symptoms, infantile adverse outcomes, and suspicion of neurodevelopment < −2SD. In the case of interaction, the analysis is stratified on effect-modifiers. Structural brain anomalies were also tested as effect-modifiers.
If an adverse outcome was not observed in one of the groups, the p value was estimated using a Fischer test (two-sided).

even when no structural brain anomalies are observed (16.7%), as compared with neonates that tested negative at birth (0.9%). At 2 years of age, infection at birth was still associated with a higher risk of neurologic impairment and/or neurosensory alteration (40.0% vs 5.2%). At 3 years of life, suspicion of neurodevelopmental delay (<−2SD) was more common in children that tested positive at birth (63.6% vs 13.7%). All those that tested positive at birth and had structural brain anomalies had evidence of neurodevelopmental delay (<−2SD) (4/4), compared with less than half in those without structural brain anomalies (3/7).

**Interpretation.** Structural brain malformations and ocular anomalies associated with congenital Zika infection have been well described worldwide, particularly in CZS[5,7,12–16]. Developmental outcomes of infants exposed to ZIKV in utero have been studied less extensively, often lacking stratification by infant ZIKV infection status at birth. The study by Nielsen-Saines and colleagues identified similar findings to us including abnormal neurodevelopmental and/or ophthalmological or auditory assessments in 31.5% of children evaluated between 7 and 32 months of age[17]. In this study, the cognitive and language domain was also the most affected (35% of 146 children). When comparing neuroimaging findings to neurodevelopmental performance in ZIKV-exposed infants, Lopes Moreira et al.[6] noted a significant association between normal results on brain imaging and higher Bayley-III scores. However, they failed to predict severe developmental delay in 2% of children and normal development in 16%. Similarly, in our cohort overall (i.e., regardless of ZIKV infection status at birth), ~20% (10/56) of the children without structural brain anomalies had evidence of neurodevelopmental delay in at least one domain at 3 years of life, whereas one-third (2/6) with brain anomalies did not. The results of the Colombian cohort reported by Mulkey et al.[18] indicate that neurodevelopmental delay in a child that is healthy at birth could worsen with age.

Brasil et al.[19] performed a study stratifying infants by their ZIKV infection status at birth. They described neurodevelopmental outcomes of 130 children born to ZIKV-infected mothers, of whom 84 (65%) tested positive for ZIKV between birth and 1 year of age. They could only observe trends towards an association between laboratory-confirmed infection and specific abnormalities (structural brain anomalies, vision and hearing deficits, abnormal neurological exam, developmental delay). The disparity between their results and ours may be explained by the difference in testing strategy for congenital infections, as a positive result after postpartum discharge is not able to differentiate congenital infections from post-natal acquired infections. As a result, a potentially higher proportion of exposure misclassification may have biased their estimates towards a null association.

The impact of the trimester of maternal infection is contradictory in some studies. In the cohort from Rio de Janeiro, the authors found that adverse outcomes were not correlated with the trimester of maternal infection[3]. Other cohorts have identified higher rates of brain structural anomalies and CZS in cases of maternal infection in the first trimester[4,16,20]. In our study, we observed a higher proportion of infants with neonatal, early infantile, or adverse outcomes at 2 and 3 years of life after maternal infection diagnosed in the first trimester of pregnancy, although this difference was not significant as our study does not appear to be sufficiently powered to conclude on this covariate (Supplementary Table 3). Moreover, our study reports the trimester at infection diagnosis but does not permit to accurately date maternal infection, as the diagnosis is based on serology in many cases.

**Relative Risks (95%CI)**

| | | |
|---|---|---|
| **Adverse outcomes at 2m** | **10.1 [3.5-29.0]** | |
| • Structural brain anomalies | 10.5 [3.0-36.7] | |
| • Severe neurologic symptoms | 26.0 [3.2-208.3] | |
| **Adverse outcomes at 2y** | **6.7 [2.2-20.0]** | |
| • Neurologic impairments | 7.1 [2.1-24.1] | |
| • Delay in motor acquisitions | 9.0 [0.8-99.6] | |
| • Neurosensory alterations | 22.5 [2.5-205.5] | |
| **Adverse outcomes at 3y** | **4.4 [1.9-10.1]** | |
| • Motor domain <-2SD | 9.3 [1.0-88.4] | |
| • Cognitive domain <-2SD | 8.3 [2.4-28.2] | |
| • Socio-affective domain<-2SD | 3.3 [1.2-9.0] | |

**Fig. 2 Main outcomes at 2 months, 2 years, and 3 years of life.** Relative risks of adverse outcomes at 2 months (m), 2 years (y), and 3 years of life, associated with laboratory-confirmed congenital Zika infection at birth are estimated using generalized linear models, adjusted on maternal infection diagnosed in the first trimester, and presented with 95% confidence intervals (95% CI). *SD* standard deviations.

**Table 3 Neonatal and early infantile outcomes, from birth to 2 months of life.**

| Neonatal and early infantile outcomes—from birth to 2 months of life | Confirmed congenital infection at birth N = 18 | Negative testing at birth N = 111 | p |
|---|---|---|---|
| *Status at 2 months of life—no (%)* | | | |
| Alive | 17 (94.4%) | 111 (100.0%) | 0.1400 |
| Neonatal demise | 1 (5.6%) | 0 (0.0%) | 0.1400 |
| *Microcephaly < −3SD* | | | |
| At birth*—no (%) | 2 (11.1%) | 1 (0.9%) | 0.0077 |
| At 2 months**—no (%)ᵃ | 2/17 (11.8%) | 1 (0.9%) | 0.0077 |
| *Weight < −2SD* | | | |
| At birth—no (%)* | 2 (11.1%) | 12 (10.8%) | 0.9256 |
| At 2 months**—no (%)ᵃ | 2/17 (11.8%) | 8/111 (7.2%) | 0.5656 |
| *Structural brain anomalies—no (%)* | 6 (33.3%) | 3 (2.7%) | <0.0001 |
| Cortical development anomaly | 4 (22.2%) | 0 (0.0%) | <0.0001 |
| Corpus callosum anomaly | 4 (22.2%) | 2 (1.8%) | 0.0030 |
| Calcifications or cystic lesions | 5 (27.8%) | 1 (0.9%) | <0.0001 |
| Posterior fossa anomaly | 4 (22.2%) | 0 (0.0%) | <0.0001 |
| Ventriculomegaly | 4 (22.2%) | 1 (0.9%) | 0.0010 |
| *Ocular anomalies—no (%)* | | | |
| Microphtalmy | 1 (5.6%) | 0 (0.0%) | 0.1400 |
| Fundoscopy anomalies | 3/10 (30.0%) | 2/34 (5.9%) | 0.0330 |
| Subretinal hemorrhage | 2/10 (20.0%) | 1/34 (2.9%) | 0.0599 |
| Chorioretinal lacunae | 2/10 (20.0%) | 1/34 (2.9%) | 0.0599 |
| Macula atrophy | 1/10 (10.0%) | 1/34 (2.9%) | 0.3462 |
| *Abnormal otoacoustic emission—no (%)* | 2/12 (16.7%) | 1/43 (2.3%) | 0.1170 |
| *Severe neurologic symptoms—no (%)* | 5 (27.8%) | 1 (0.9%) | <0.0001 |
| Arthrogryposis | 1 (5.6%) | 0 (0.0%) | 0.1400 |
| Hypertonia | 3 (16.7%) | 0 (0.0%) | 0.0020 |
| Dysphagia/swallowing disorders | 2 (11.8%) | 0 (0.0%) | 0.0190 |
| Seizures | 1 (5.6%) | 1 (0.9%) | 0.2610 |
| *NICU admission—no (%)* | 3 (16.7%) | 14 (12.6%) | 0.6372 |

Weight and head circumference (HC) variables are presented as medians with extreme values. For qualitative variables, absolute frequencies and relative frequencies (percentages) are presented. Secondary outcomes were compared across the groups using $\chi^2$, Fischer, and Wilcoxon tests.
ᵃ128 alive infants evaluated at 1 and 2 months of life.
*According to Intergrowth21 charts.
**According to WHO Child Growth Standards. Exact p values were estimated by $\chi^2$, Fischer, or Wilcoxon tests (two-sided).

Overall, the results from our study along with those from previously published studies seem to indicate that a laboratory-confirmed congenital ZIKV infection at birth could be associated with higher risks of long-term outcomes, even in children without structural brain anomalies. As a normal antenatal and neonatal evaluation cannot provide complete reassurance for children exposed to ZIKV in utero, it seems paramount to offer systematic testing for congenital ZIKV infection at birth in cases of in utero exposure and to adapt counseling according to these results.

**Study limitations.** The first limitation of this study is the proportion of individuals lost to follow-up reducing the sample size from 129 to 111 after 2 years and to 62 after 3 years and introducing a potential selection bias. Loss to follow-up is critical in determining a study's validity as patients lost to follow-up might have a different outcome than those who complete the study. In our study, although not significant, the proportion of loss to follow-up was higher among children who tested negative at birth compared with those with a confirmed congenital infection (60/111, 54.1% vs 6/17, 35.3%, $p = 0.1494$, Std diff = 0.38), which suggests a potential selection bias on the outcome. Yet, it is difficult to know if the loss to follow-up has selected the more severe cases or not. One would argue that the lack of clinical concern by parents, particularly in asymptomatic cases, might have driven the loss to follow-up. This would have overestimated the absolute risks of infantile adverse outcomes and the suspicion of neuro-development delay in the cohort, particularly in those that tested negative at birth. Thus, absolute risk in this study should be considered carefully.

Another source of potential selection bias is linked to practical limitations for follow-up at the CHOG for newborns from mothers living along the Maroni River or in isolated areas in Suriname. Among the newborns born at the CHOG, only those that would be followed at the CHOG pediatric clinic were enrolled, resulting in the exclusion of 35% of the CHOG-born newborns from ZIKV-infected mothers, leading to the initial inclusion of only 129 infants. This might have selected infants stemming from families with a higher socioeconomic status. This would have led to a possible underestimation of the absolute risk although unlikely as this selection impacted both groups.

The second limitation of this study is the testing performance to confirm congenital infections. In fetuses and neonates, it has been demonstrated that viremia is transient in blood, amniotic fluid, and urine[21]. Thus, the window to detect congenital infections using reverse transcription-polymerase chain reaction (RT-PCR) may be shorter than for other congenital infections (i.e., CMV). In infants, congenital ZIKV infections are difficult to confirm retrospectively, owing to serological test cross-reaction and the possibility of infection after birth in the context of continuous exposure. To avoid false-negative or false-positive results, neonatal serology was performed before postpartum discharge. As no other flaviviridae was circulating significantly during this period, we considered the risk of cross-reactions low, and a positive neonatal IgM without positive RT-PCR was considered as a laboratory-confirmed congenital infection, although these would be considered as probable cases per CDC definitions[22]. We tried to reduce the risk of misclassification biases by performing neonatal testing on different samples (Appendix 1), however, we cannot exclude that some newborns

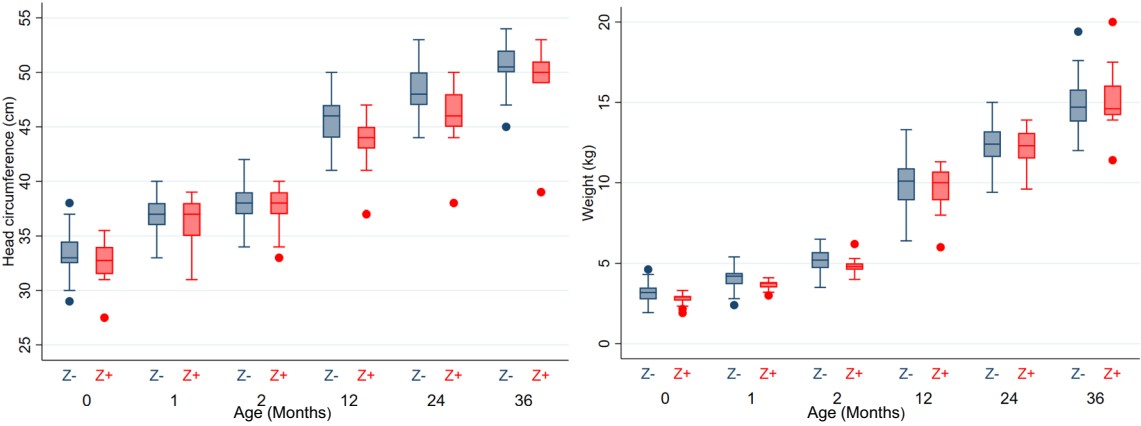

**Fig. 3 Head circumference and weight in children tested positive and negative for congenital Zika infection at birth.** Head circumferences (in centimeters) and weights (in kilograms) are presented for each time point of this study (birth, 1 month, 2 months, 1 year, 2 years, 3 years). Boxes represent median and interquartile range (IQR), whiskers represent range excluding outliers >1.5× IQR from upper or lower quartile, and circles represent outliers. Z- (blue): Children tested negative at birth for congenital Zika infection; Z+ (red): children tested positive at birth for congenital Zika infection. n = 129 neonates at birth (18 Z+/ 111 Z−), n = 128 infants at 1 and 2 months (17 Z+/ 111 Z−), n = 112 infants at 12 months (15 Z+/ 97 Z−), n = 111 children at 24 months (15 Z+/ 96 Z−), n = 62 children at 36 months (11 Z+/51 Z−).

**Table 4 Outcomes up to 3 years of life.**

| Children outcomes—up to 3 years of life | Confirmed congenital infection at birth N = 15 | Negative testing at birth N = 97 | p |
|---|---|---|---|
| *Microcephaly < −3SD** | | | |
| At 1 year | 2 (13.3%) | 1 (1.0%) | 0.0060 |
| At 2 years[a] | 2 (13.3%) | 1/96 (1.0%) | 0.0080 |
| At 3 years[b] | 2/11 (9.1%) | 1/51 (2.0%) | 0.0237 |
| *Weight < 5th percentile** | | | |
| At 1 years | 2 (13.3%) | 6 (6.2%) | 0.2908 |
| At 2 years[a] | 2 (13.3%) | 6/96 (6.3%) | 0.2947 |
| At 3 years[b] | 1/11 (9.1%) | 3/51 (5.9%) | 0.6944 |
| *Neurologic impairments at 2y[a]—no (%)* | 5 (33.3%) | 4/96 (4.2%) | 0.0001 |
| Cerebral palsy | 2 (13.3%) | 0/96 (0.0%) | 0.0170 |
| Severe dystonia or tremors | 3 (20.0%) | 3/96 (3.1%) | 0.0070 |
| Seizures | 3 (20.0%) | 2/96 (2.1%) | 0.0170 |
| *Motor acquisitions* | | | |
| Age at sitting position (m)—median (min-max) | 6 (3–24) | 6 (4–11) | 0.8113 |
| Delay for sitting position (>9 m)-no (%) | 2 (13.3%) | 1 (1.0%) | 0.0060 |
| Age at walking (m)—median (min-max) | 11 (8–24) | 11 (7–17) | 0.3289 |
| Delay for walking[a] (>18 m)—no (%) | 1 (6.7%) | 0/96 (0.0%) | 0.0110 |
| *Vision and hearing evaluation* | | | |
| Impaired response to visual stimuli—no (%) | 2 (13.3%) | 1 (1.0%) | 0.0060 |
| Impaired response to auditory stimuli—no (%) | 2 (13.3%) | 1 (1.0%) | 0.0060 |
| *Age at CDAS evaluation[b] (m)—median (min-max)* | 35 (33–39) | 36 (34–40) | 0.7665 |
| Global assessment - no (%) | | | |
| "Comfort" zone (>−1SD) | 3/11 (27.3%) | 30/51 (58.8%) | 0.0572 |
| "To be monitored" zone ([−2SD; −1SD]) | 1/11 (9.1%) | 14/51 (27.5%) | 0.1972 |
| "Referral" zone (<−2SD) | 7/11 (63.4%) | 7/51 (13.7%) | 0.0003 |
| Motor domain—no (%) | | | |
| "Comfort" zone (>−1SD) | 7/11 (63.4%) | 47/51 (92.2%) | 0.0105 |
| "To be monitored" zone ([−2SD; −1SD]) | 2/11 (18.2%) | 3/51 (5.8%) | 0.1742 |
| "Referral" zone (<−2SD) | 2/11 (18.2%) | 1/51 (2.0%) | 0.0790 |
| Socio-emotional domain—no (%) | | | |
| "Comfort" zone (>−1SD) | 7/11 (63.4%) | 41/51 (80.4%) | 0.2280 |
| "To be monitored" zone ([−2SD; −1SD]) | 0/11 (0.0%) | 4/51 (7.8%) | 0.3369 |
| "Referral" zone (<−2SD) | 4/11 (36.4%) | 6/51 (11.8%) | 0.0442 |
| Cognitive and language domain—no (%) | | | |
| "Comfort" zone (>−1SD) | 3/11 (27.3%) | 32/51 (62.7%) | 0.3140 |
| "To be monitored" zone ([−2SD; −1SD]) | 2/11 (18.2%) | 16/51 (31.4%) | 0.3820 |
| "Referral" zone (<−2SD) | 6/11 (54.5%) | 3/51 (5.9%) | <0.0001 |

For qualitative variables, absolute frequencies and relative frequencies (percentages) are presented. Secondary outcomes were compared across the groups using $\chi^2$, Fischer, and Wilcoxon tests.
[a]111 infants evaluated at 2 years of life, including 15 with laboratory-confirmed congenital ZIKV infection.
[b]62 children evaluated at 3 years of life using the CDAS, including 11 with laboratory-confirmed congenital ZIKV infection.
*According to WHO Child Growth Standards <2 years and CDC growth charts >2 years of life. Exact p values were estimated by $\chi^2$, Fischer, or Wilcoxon tests (two-sided).

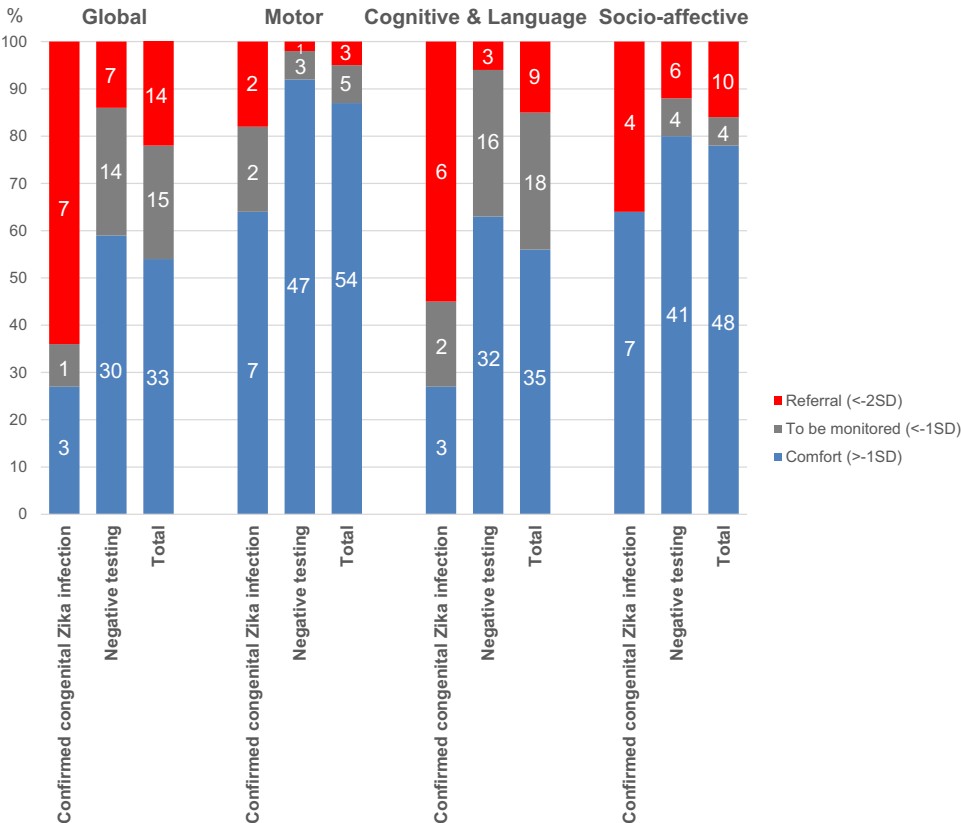

**Fig. 4 Childhood development at 3 years of life.** All children followed for in utero ZIKV exposure were recalled for a developmental evaluation using the Child Development Assessment Scale at 3 years of life (33–42 months, n = 62). Normal results are classified in the "comfort" or "blue" zone (>−1 standard deviation [SD]). Intermediate results are classified in the "to be monitored" or "gray" zone (−2SD; −1SD). Suspicion of delays is classified in the "referral" or "red" zone (<−2SD). The motor, socio-emotional, and cognitive & language domains were evaluated using this scale. Results of these domains were synthesized in a global evaluation.

classified as uninfected had undetectable viremia and immune response at birth. To increase the sensitivity of neonatal testing, we included positive RT-PCR on placental samples in the definition of laboratory-confirmed congenital ZIKV infections[23]. However, we did not observe contradictory results in cases of infected placentas, as all also had a positive IgM and/or RT-PCR in fetal/neonatal samples.

The follow-up of infants and children was based on the French recommendations[24] and adapted to local capacities, but we cannot exclude that routine MRI, auditory brainstem response testing, and consultation with a pediatric neurologist, as recommended by the CDC, would have diagnosed more subtle and specific signs of congenital ZIKV infection.

The third limitation is the presence of a language barrier. Some children or mothers have difficulties when using the French language. These difficulties could have wrongly led to a lower score when using the French version of the CDAS to evaluate the cognitive & language domain, resulting in misclassification of the outcome. The two practitioners who evaluated these children, however, were also able to speak the local language and translate questions, limiting the poor understanding of the CDAS assessment. This would have led to an overestimation of the difficulties in the cognitive and language domain, however, this issue would likely have impacted both groups equally.

The last limitation was that a control group of children born from uninfected mothers who underwent neurodevelopmental testing using the CDAS was not available. In the general population, a normal distribution of neurodevelopmental scores would be expected when using a standardized tool such as the

CDAS, but this test has never been used in French Guiana and cognitive scales, in particular, may include items that could be influenced by the cultural context. In a cross-sectional study evaluating the neurodevelopment of Polynesian infants born during the ZIKV outbreak versus a control group of Canadian infants, Subissi et al.[25] described that confounding factors such as socioeconomic status and cultural factors may play an important role in infantile neurodevelopmental assessment.

## Methods

**Study settings and participants**. This prospective cohort study included new-borns from mothers infected with ZIKV during pregnancy that were followed at the CHOG after the 2016 ZIKV epidemic. The CHOG offers the only maternity and neonatal intensive care units in western French Guiana. During the ZIKV epidemic (January to September 2016), all pregnant women in the territory underwent laboratory screening by ZIKV serology in each trimester and at delivery, as well as RT-PCR in urine and plasma samples for those with symptoms.

All infected women were followed in the fetal medicine unit of the CHOG. Fetal ultrasound (US) examinations were performed every 3–4 weeks using E8 and E10 Voluson scanners with abdominal (RM6C) and transvaginal (RIC5-9-D) transducers (General Electric Healthcare, Zipf, Austria). Additional investigations (MRI, computed tomography (CT), amniocentesis) were performed based on US results and after a discussion with a multi-disciplinary team. All neonates from ZIKV-infected mothers living in the area (Fig. 5) were offered ongoing follow-up at the CHOG until the third year of life and participation in this study. Asymptomatic neonates from mothers living along the Maroni River, outside of the Saint-Laurent du Maroni area, were discharged with their mother after day 3–5 postpartum and were followed in the nearest primary healthcare center, and only returned to the CHOG in cases of emergency or need for advanced care. Thus, these infants were not included in this cohort.

The study received ethics approval from the institutional review board of the CHOG and written consent of the mother was obtained.

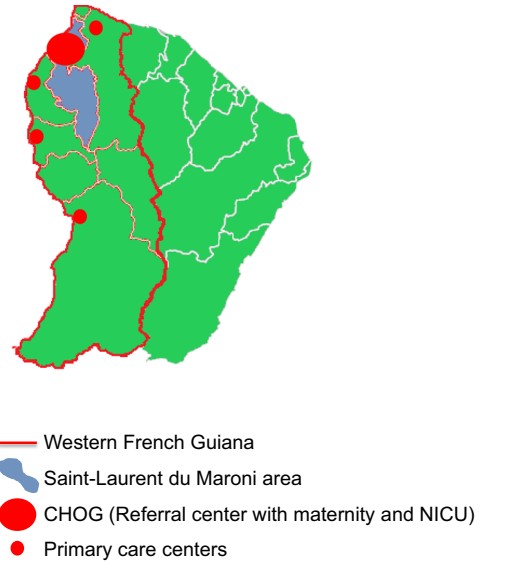

**Fig. 5 Healthcare services in western French Guiana.** All newborns from Zika-infected mothers, living in western French Guiana (within the red lines) and followed at the pediatric clinic of the CHOG (large red dot), were enrolled in this cohort. Asymptomatic neonates from mothers living outside of the Saint-Laurent du Maroni area (blue area), were discharged after day 3–5 postpartum and were followed in the nearest primary healthcare center (small red dot).

Western French Guiana
Saint-Laurent du Maroni area
CHOG (Referral center with maternity and NICU)
Primary care centers

**Laboratory testing for congenital ZIKV infection and exposure definition**. During pregnancy, RT-PCR on amniotic fluid was offered for cases with fetal anomalies or if an amniocentesis was performed for another indication (i.e., aneuploidy diagnosis). After birth, all newborns underwent ZIKV serology for detection of specific IgM before day 3 of life. RT-PCRs were performed in cord blood, neonatal urine, and placenta. Additional testing on cerebrospinal fluid was proposed in cases with neurological symptoms or demise.

We defined a laboratory-confirmed congenital ZIKV infection either by positive RT-PCR from at least one fetal/neonatal sample (amniotic fluid, cerebrospinal fluid, urine, blood, placenta) or identification of specific IgM in neonatal blood or in cerebrospinal fluid.

Neonates from ZIKV-infected mothers without a confirmed congenital ZIKV infection were classified as controls.

Molecular and serologic testing was performed at the French Guiana National Reference Center for arboviruses (Institut Pasteur of French Guiana, Cayenne, French Guiana) using the Realstar Zika Kit (Altona Diagnostics GmbH, https://altona-diagnostics.com) for RT-PCR, and in-house enzyme-linked immunosorbent assay (ELISA) and IgM antibody-capture (MAC) ELISA for serologic testing. The limit of detection for serum samples tested using the Realstar Zika Kit was 0.61 (95% CI 0.39–1.27) copies/µL, and a cycle threshold value <37 was considered positive. The following primers were used: (1) ZIKV 1086 1086–1102 CCGCTG CCCAACACAAG; (2) ZIKV 1162c 1162–1139 CCACTAACGTTCTTTTGCAG ACAT; (3) ZIKV 1107-FAM 1107–1137 AGCCTACCTTGACAAGCAGTCAGA CACTCAA. ELISA and MAC-ELISA testing was based on whole virus antigens, with a positive result defined as three standard deviations above the negative control value. Its sensitivity for specific IgM detection was estimated at 98% after day 7 from symptoms onset in an adult cohort[26].

Details of maternal, fetal, and neonatal testing are available in our previous studies[4,11]. Placentas were sampled and tested according to the method described in our dedicated study, which also corresponds to what was proposed by Seferovic and colleagues[23,27].

**Outcome definition and time of measurement**. Newborns underwent cerebral imaging and neurosensory testing and were followed by a pediatrician up to 3 years of life. The last evaluation included a neurodevelopmental screening using the CDAS[28].

**Neonatal and early infantile outcomes**. All ZIKV-exposed neonates, regardless of their testing result at birth, underwent clinical examination with special attention to anthropometric measurements, neurological status, and signs of infection. The HC measurements were confirmed 24 h after birth to avoid the effects of delivery sequelae. In addition to clinical examination, they were assessed by transfontanellar US, hearing evaluation by OAE testing, and fundoscopy (1–2 months after birth). Every abnormal examination was reconfirmed, and more investigations (MRI, CT) were requested depending on the clinical picture.

Neonatal and early infantile adverse outcomes were defined as neonatal death (between birth and 2 months of life, intrapartum demise not included), structural brain anomalies, or severe neurological symptoms (according to Pomar et al., BMJ, 2018; Prenat diagnosis, 2019)[11,29].

**Follow-up and outcomes at 2 years of life**. All infants enrolled were scheduled for medical consultation at the CHOG pediatric clinic at 2, 6, 9, 12, 18, and 24 months of life. These pediatric examinations included parental questioning on infant development, anthropometric measurements, assessment of motor acquisition, a neurological examination, and an auditory and visual assessment, following the French high Council of Public Health (HCSP) recommendations[24].

Adverse outcomes at 2 years of life were defined as the observation of neurologic impairment (cerebral palsy, severe dystonia, tremors, or seizures), motor acquisition delay (sitting position >9 months or walking >18 months of age), or neurosensory alterations (impaired response to visual or auditory stimuli) until the age of 24 months.

**Neurodevelopmental outcomes**. In August and September 2019, at 3 years of life, the children were screened for neurocognitive development using the French version of the CDAS[28]. Adapted to children 0–5 years of age, the results allow the user to evaluate the child's cognitive, language, motor, and social-emotional development using a validated and standardized scale. Results in the "comfort zone" (blue, >−1SD) indicate normal development. Results in the "to be monitored zone" (gray, [−2SD to −1SD]) suggest that interventions with the child should be adapted according to identified difficulties and that the child should be reassessed later. Finally, results in the "referral zone" (red, <−2SD) indicate that the child should be referred for an exhaustive developmental assessment. To avoid any bias of administration, all children were evaluated by two medical doctors trained to perform the CDAS test and blinded for the results of congenital Zika infection testing.

A suspected delay in neurodevelopment was defined as a CDAS below −2SD ("referral zone") in at least one domain at 3 years of life.

**Mitigation of bias**. In the context of French Guiana, infants have an increased risk of loss to follow-up after postpartum discharge, as some live in isolated areas and are followed in primary care centers. These infants were not included in the cohort to avoid misclassification bias. In infants enrolled in the cohort, when missed appointments occurred, the parents were recalled to schedule another evaluation. We did not enroll infants referred to the CHOG for advanced care who were not included initially in the cohort, to avoid selection biases.

OAE testing was implemented in May 2016 and infants born before that time were not systematically tested at birth. After the epidemic peak, we encountered human and technical limitations to perform fundoscopy in all children born from ZIKV-positive mothers. To avoid selection and classification biases, we did not consider abnormal fundoscopies or OAE in primary outcomes. Instead, we considered an abnormal response to auditory or visual stimuli in infancy, as all the infants were tested for these outcomes.

**Statistical analysis**. Baseline characteristics of mothers and newborns were obtained at enrollment and presented as absolute and relative frequencies for those diagnosed with a laboratory-confirmed congenital Zika infection at birth and those that tested negative. Timing of maternal infection diagnosis was estimated based on symptom onset or on laboratory results in cases of asymptomatic infection; and grouped into first or second and third trimesters for the analysis. Gestational age at birth was considered as a binary variable for the analysis ("prematurity < 37 wg").

Standardized differences were calculated to compare baseline characteristics of patients with laboratory-confirmed congenital ZIKV infection at birth to those who tested negative. These characteristics were considered unbalanced when the standardized difference was >0.1.

The relative risks (RR) associated with laboratory-confirmed congenital ZIKV infection were assessed using generalized linear models, and were adjusted (aRR) for confounding factors (trimester at maternal ZIKV infection diagnosis), and controlled for potential interactions with exposures during pregnancy, maternal age, co-morbidities, and socioeconomic status, infant gender, twins, prematurity and the mode of delivery. Structural brain anomalies were also tested as effect-modifiers for severe neurological symptoms, adverse outcomes at 2 years of life, and suspicion of neurodevelopmental delay < −2SD. In the case of interaction, the analysis was stratified for effect-modifiers.

We performed a complete case analysis, thus using different denominators for outcomes at 2 months, 2 years, and 3 years of life.

Data were collected using Excel software and analyzed using Stata 15 (Stata Corporation, College Station, TX, USA).

**Reporting summary**. Further information on research design is available in the Nature Research Reporting Summary linked to this article.

## Data availability
Source data that underlie the results are provided with this paper (supplementary information). Other individual participant data will be shared with researchers who

provide a methodologically sound proposal for the multicentric study, particularly individual participant data meta-analysis. Proposals should be directed to leo. pomar@chuv.ch. Source data are provided with this paper.

## Code availability

The codes used in Stata 15 software to clean, analyze and present Tables 1–4 are provided with this paper ("Supplementary Codes").

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

## Author contributions

N.H. conceived and designed the study, provided care to mothers and children, collected the data, and draft the initial manuscript. Y.K. and Z.H.L.R. provided care to children, collected data, and reviewed and revised the manuscript. V.L., M.M., and G.C. participated in the study design, provided care to mothers, collected data, and reviewed and revised the manuscript. D.B. and A.P. analyzed and interpreted the data, and reviewed and revised the manuscript. L.P. conceived and designed the study, provided care to mothers, analyzed and interpreted the data, and drafted the initial manuscript. All authors approved the final manuscript as submitted and agree to be accountable for all aspects of the work.

## Competing interests

The authors declare no competing interests.
