## [Peer Review File · Nature Communications]

REVIEWER COMMENTS

Reviewer #1 (Remarks to the Author):

I wish to thank the authors and editors for the opportunity to review this very important body of work. While it has considerable challenges in its present form, the data and information contained is exceedingly important for medicine, science and public policy. It is a unique cohort and dataset and the information will be of high impact and high yield.

Several key issues:

1. A real strength is the authors use of placental testing to determine positivity at birth, not just neonatal serologies or PCR. This was previously reported in PMID: 30736425 by another group, and is worth highlighting again here.

That said, precise definitions of positive and negative need to be included. What IgM titer positive? What Ct value by RT PCR? Also, they actually leave out placenta positive in their case definitions. This section is crucially important and must be unequivocally clear.

This similarly applies to all of their testing and measurements of outcomes. Either reference the method used or provide a description that is clear, unequivocal, and others can use. This is important, because others wishing to implement the testing performed in this cohort needs the very, very clear clinical tools and measures used.

2. Their terminology is not used in standard accepted from. A neonate, by definition, is 1-30 days of life, and infant 31 days to 1 year, and subsequently designated by years. Use the correct terminology...neonates are not 2 months of age. This means that they either need to reanalyze their data or correctly relabel their analysis groups.

3. Their handling of selection bias is commendable, and a real strength of the study. However, reporting percentages and numbers without any measure of significance of difference (Table 1, 3, 4, 5) is not robust. They could report by t-test, or more appropriately OR and adjusted ORs. They also likely want to employ sensitivity testing and propensity scores, as well as Cox proportional hazard ratios since their time of follow up was distinct.

For the aORs, they need to control for confounders that are actually different in their cases and controls, not make guesses. I might suggest consultation with an epidemiologist/biostatistician.

Several tables could be alternately projected as forest plots with ORs.

4. There are many opportunities for editing the text and syntax.

Reviewer #2 (Remarks to the Author):

The manuscript by Heini and colleagues presents the results from a prospective cohort study of 129 infants born of zika-infected mothers in French Guiana. Evaluations are conducted at 4 timepoints: birth, 2 months, 2 years, 3 years.

The selection procedures for enrolment in the cohort and follow-up are clearly described and the flowchart (figure 1) helps understanding the sample size at the different timepoints.

The statistical methods used are adapted and well presented. Statistical comparison of baseline characteristics, between zika-infected and zika-non-infected children at birth, should however be conducted and the tests chosen mentioned in the statistical analysis section. I would recommend chi-2 tests for the categorical variables and non-parametric Wilcoxon tests for continuous variables. Thus, p-values could be added in Table 1 and in the results section. Indeed, in the results section, the authors compared proportions between infected and non-infected children, but without p-values it is not intuitive to identify whether there are statistical differences or not.

Because the number of children for the evaluations at 2 months, 2 years and 3 years is not constant, I would recommend to start each paragraph in the results section but stating the sample size. The authors attempted to do so, but it was not systematic.

In Table 2, it is unclear how the authors could estimate p-values in the sub-analysis in neonates with structural brain anomalies as no neonate adverse outcome was observed in non-infected children, and they do not report a relative risk. The same way for demise. If the authors performed a chi-2 test, this should be reported. The low number of children with structural brain anomalies makes it difficult to evidence statistical differences; and as a consequence, lack of statistical difference may only be due to the small sample size. I wonder if presenting this sub-analysis is relevant.

In Table 3 and 4, p-values would be helpful to better apprehend the differences between the infected and non-infected children. This would be helpful as results presented in tables 3 and 4 are hardly mentioned in the results section and are not really discussed.

The statistical analyses were adjusted for co-factors, but the effect of some cofactors would have been of interest to the readers, especially the timing of the infection during pregnancy in the mothers.

In the statistical analysis section, the authors mentioned that stratified analysis would be performed in case interaction was evidenced. However, in the results section, no reference is made to interactions. Were interactions tested but found to be not significant? If this is the case, I would recommend mentioning it.

The conclusions are supported by the results presented, but I would have liked the effect of the timing of the infection during pregnancy in the mothers to be discussed.

The discussion about lost to follow-up patients is interesting. However, the authors state that the proportions are similar in both groups, but no p-values is provided to convince the readers that this is true. And it feels like the proportion of lost to follow-up is higher in non-infected children. Again, giving p-values would convince that the samples are not biased due to a higher proportion of lost to follow-up in a group.

In conclusion, the authors provide valuable data, analysed using adequate statistical methods, on the evolution of children exposed to zika virus during pregnancy and compare the evolution based on their infection status at birth.

REVIEWER COMMENTS

Reviewer #1 (Remarks to the Author):

I wish to thank the authors and editors for the opportunity to review this very important body of work. While it has considerable challenges in its present form, the data and information contained is exceedingly important for medicine, science and public policy. It is a unique cohort and dataset and the information will be of high impact and high yield.

We thank the reviewer for their very positive comment and for their help to improve our manuscript.

Several key issues:

1. A real strength is the authors use of placental testing to determine positivity at birth, not just neonatal serologies or PCR. This was previously reported in PMID: 30736425 by another group, and is worth highlighting again here.

That said, precise definitions of positive and negative need to be included. What IgM titer positive? What Ct value by RT-PCR? Also, they actually leave out placenta positive in their case definitions. This section is crucially important and must be unequivocally clear.

This similarly applies to all of their testing and measurements of outcomes. Either reference the method used or provide a description that is clear, unequivocal, and others can use. This is important, because others wishing to implement the testing performed in this cohort needs the very, very clear clinical tools and measures used.

Our neonatal testing indeed went beyond CDC recommendations by using more samples, especially placental samples. We would like to thank the reviewer for the parallels made with the study of Seferovic and colleagues, and we have integrated this reference into our methods. The positivity of placental specimens has been added to the definition of laboratory-confirmed congenital infection. This change does not compromise our results, as all newborns with a positive placenta also had other positive results (IgM or RT-PCR in blood, urine, amniotic fluid or cerebro-spinal fluid, see Appendix 1).

We have included the methods and the thresholds used for molecular and serologic testing in this section:

“During pregnancy, RT-PCR on amniotic fluid was offered for cases with fetal anomalies or if an amniocentesis was performed for another indication (i.e. aneuploidy diagnosis). After birth, all newborns underwent ZIKV serology for detection of specific IgM before day three of life. RT-PCRs were performed in cord blood, neonatal urine and placenta. Additional testing on cerebrospinal fluid was proposed in cases with neurological symptoms or demise.

We defined a laboratory confirmed congenital ZIKV infection either by positive RT-PCR from at least one fetal/neonatal sample (amniotic fluid, cerebrospinal fluid, urine, blood, placenta) or identification of specific IgM in neonatal blood or in cerebrospinal fluid.

Neonates from ZIKV-infected mother without a confirmed congenital ZIKV infection were classified as controls.

Molecular and serologic testing was performed at the French Guiana National Reference Center for arboviruses (Institut Pasteur of French Guiana, Cayenne, French Guiana) using the Realstar Zika Kit (Altona Diagnostics GmbH, <https://altona-diagnostics.com>) for RT-PCR, and in-house IgM and IgG antibody-capture ELISA for serologic testing. The limit of detection for serum samples tested using the Realstar Zika Kit was 0.61 (95% CI 0.39–1.27) copies/ μ L, and a cycle threshold value <37 was considered positive. MAC-ELISA testing was based on whole virus antigens, with a positive result defined as three standard deviations above the negative control value. Its sensitivity for specific IgM detection was estimated at 98% after day 7 from symptom onset in an adult cohort²⁵. Details of maternal, fetal and neonatal testing are available in our previous studies^{4,11}. Placentas were sampled and tested according to the method described in our dedicated study, which also corresponds to what was proposed by Seferovic and colleagues^{24,25}”.

In the discussion, we have mentioned that we did not observe cases with positive placenta without another positive sample:

“The second limitation of this study is the testing performance to confirm congenital infections. In fetuses and neonates it has been demonstrated that viremia is transient in blood, amniotic fluid and urine²⁰. Thus, the window to detect congenital infections using RT-PCR may be shorter than for other congenital infections (i.e. CMV). In infants, congenital ZIKV infections are difficult to confirm retrospectively, due to serological test cross-reaction and the possibility of an infection after birth in the context of a continuous exposure. To avoid false negative or false positive results, neonatal serology was performed before postpartum discharge. As no other flaviviridae was circulating significantly during this period, we considered the risk of cross-reactions low, and a positive neonatal IgM without positive RT-PCR was considered as a laboratory confirmed congenital infection, although these would be considered as probable cases per CDC definitions²¹. We tried to reduce the risk of misclassification biases by performing neonatal testing on different samples (Appendix 1), however we cannot exclude that some newborns classified as uninfected had undetectable viremia and immune response at birth. To increase the sensitivity of neonatal testing, we included positive RT-PCR on placental samples in the definition of laboratory-confirmed congenital ZIKV infections²². However, we did not observe contradictory results in cases of infected placentas, as all also had a positive IgM and/or RT-PCR in fetal/neonatal samples.”

2. Their terminology is not used in standard accepted from. A neonate, by definition, is 1-30 days of life, and infant 31 days to 1 year, and subsequently designated by years. Use the correct terminology...neonates are not 2 months of age. This means that they either need to reanalyze their data or correctly relabel their analysis groups.

We agree that the terminology used can be confusing. We have chosen to relabel our analysis groups to clarify the timing of the outcomes. We have modified the abstract, introduction, methods, results, discussion, and tables to reflect this change.

Outcomes up to two months of life are now defined as “Neonatal and early infantile outcomes.” Later outcomes are labeled as “Outcomes at two years of life” and “Neurodevelopment at three years of life”

3. Their handling of selection bias is commendable, and a real strength of the study. However, reporting percentages and numbers without any measure of significance of difference (Table 1, 3, 4, 5) is not robust. They could report by t-test, or more appropriately OR and adjusted ORs. They also likely want to employ sensitivity testing and propensity scores, as well as Cox proportional hazard ratios since their time of follow up was distinct.

For the aORs, they need to control for confounders that are actually different in their cases and controls, not make guesses. I might suggest consultation with an epidemiologist/biostatistician.

Several tables could be alternately projected as forest plots with ORs.

We did not perform association tests on tables 1, 3, 4 and 5 because they present the baseline characteristics and secondary outcomes, and measuring associations not scheduled in a data management plan, especially for baseline characteristics, is contrary to current recommendations (American Statistical Association, 2019; Amrhein, Nature, 2019; Harrington, NEJM, 2019). We opted to measure the balance between these two groups by standardized differences, which are much more appropriate for small groups (Flury, The American Statistician, 1986). Baseline characteristics were considered unbalanced when standardized differences were greater than 0.1 (as suggested by Normand, J Clin Epidemiol, 2001). We observed standardized differences greater than 0.1 for maternal age, exposures during pregnancy, trimester of maternal infection and mode of delivery. Relative risks for the main outcomes were adjusted by the trimester of maternal infection (considered as a confounding factor as the risk of trans-placental infection and potential fetal consequences might differ between 1st and 3rd trimester infection (Ades, Lancet id, 2020)). The other unbalanced characteristics were tested as effect-modifiers, as it is specified in the methods.

We acknowledge that standardized differences were not presented in our original manuscript, and we have therefore included them in the methods and Table 1.

- Methods: “Standardized differences were calculated to compare baseline characteristics of patients with laboratory-confirmed congenital ZIKV infection at birth to those who tested negative. These characteristics were considered unbalanced when the standardized difference was >0.1.
The Relative Risks (RR) associated with laboratory confirmed congenital ZIKV infection were assessed using generalized linear models, and were adjusted (aRR) for confounding factors (trimester of maternal ZIKV infection), and controlled for potential interactions with exposures during pregnancy, maternal age, co-morbidities and socio-economic status, infant gender, twins, prematurity and the mode of delivery. Structural brain anomalies were also tested as effect-modifiers for severe neurological symptoms, infantile adverse outcomes at 2 years of life and suspicion of neurodevelopmental delay <-2SD. In case of interaction, the analysis was stratified for effect-modifiers.”

- Table 1:

Characteristics	Confirmed congenital infection at birth N= 18	Negative testing at birth N= 111	Std Diff
Maternal age at birth (years) - median (min-max)	25 (18-38)	26 (18-43)	0.09
Maternal socio-economic status – no. (%)			
Low	6 (33.3%)	38 (34.2%)	-0.02
Moderate	8 (44.4%)	46 (41.4%)	0.06
High	1 (5.6%)	8 (7.2%)	-0.07
Unknown	3 (16.7%)	19 (17.1%)	-0.01
Maternal exposure during pregnancy – no. (%)			
Alcohol consumption	1 (5.6%)	12 (10.8%)	-0.19
Drug use	0 (0.0%)	1 (0.9%)	0.13
Current smoker	1 (5.6%)	5 (4.5%)	0.05
Lead poisoning	1 (5.6%)	6 (5.4%)	0.01
Any maternal comorbidities ^π – no (%)	3 (16.7%)	15 (13.5%)	0.09
Diabetes (previous or gestational)	1 (5.6%)	7 (6.3%)	-0.03
Vascular pathologies	1 (5.6%)	5 (4.5%)	0.05
Severe anemia	1 (5.6%)	5 (4.5%)	0.05
Co-infections*	1 (5.6%)	2 (1.8%)	0.20
Maternal Zika infection – no (%)			
Symptomatic	4 (22.2%)	22 (19.8%)	0.06
Asymptomatic	14 (77.8%)	89 (80.2%)	-0.06
Trimester of Maternal Zika infection – no. (%)			
T1 infection	7 (38.9%)	27 (24.3%)	0.31
T2 infection	7 (38.9%)	37 (33.3%)	0.11
T3 infection	2 (11.1%)	25 (22.5%)	-0.30
Unknown	2 (11.1%)	22 (19.8%)	-0.24
Dichorionic twins	2 (11.1%)	6 (5.4%)	0.20
Fetus gender – no. (%)			
Female	10 (55.6%)	63 (56.8%)	-0.02
Male	8 (44.4%)	48 (43.2%)	0.02
Term at birth (weeks' gestation) - median (min-max)	39 (32-41)	39 (36-41)	-0.18
Premature birth <37wg	1 (5.6%)	6 (5.4%)	0.01
Mode of delivery – no. (%)			
Normal	13 (72.2%)	97 (87.4%)	-0.38
C-section	5 (27.8%)	14 (12.6%)	0.38
Neonatal adaptation			
Abnormal Apgar score (<6 at 5 min) – no (%)	2 (11.1%)	13 (11.7%)	-0.02
Abnormal Lactate level (> 4.5) – no (%)	2 (11.1%)	16 (14.4%)	-0.09

^π including multiple maternal co-morbidities

* one primary cytomegalovirus infection in the infected group; one primary CMV and one primary toxoplasmosis

Alcohol consumption was defined as ongoing consumption during the pregnancy after its diagnosis

Current smoking was defined as ongoing smoking during pregnancy after its diagnosis

Trimester of maternal Zika infection was estimated based on symptoms onset, or on laboratory results (asymptomatic)

Baseline characteristics were considered unbalanced if standardized differences (Std diff) were >0.1

We, however, recognize that different ways of exploring potential confounding factors are possible, and in following the advice of the two reviewers, we present p-values from Chi-2, Fisher and Wilcoxon tests in Table 1 (as suggested by Reviewer 2):

Characteristics	Confirmed congenital infection at birth N= 18	Negative testing at birth N= 111	p
Maternal age at birth (years) - median (min-max)	25 (18-38)	26 (18-43)	0.5895
Maternal socio-economic status – no. (%)			
Low	6 (33.3%)	38 (34.2%)	0.9400
Moderate	8 (44.4%)	46 (41.4%)	0.8110
High	1 (5.6%)	8 (7.2%)	0.6340
Unknown	3 (16.7%)	19 (17.1%)	0.9620
Maternal exposure during pregnancy – no. (%)			
Alcohol consumption	1 (5.6%)	12 (10.8%)	0.4290
Drug use	0 (0.0%)	1 (0.9%)	0.8600
Current smoker	1 (5.6%)	5 (4.5%)	0.6050
Lead poisoning	1 (5.6%)	6 (5.4%)	0.6600
Any maternal comorbidities [‡] – no (%)	3 (16.7%)	15 (13.5%)	0.4770
Diabetes (previous or gestational)	1 (5.6%)	7 (6.3%)	0.6910
Vascular pathologies	1 (5.6%)	5 (4.5%)	0.6020
Severe anemia	1 (5.6%)	5 (4.5%)	0.6020
Co-infections*	1 (5.6%)	2 (1.8%)	0.3650
Maternal Zika infection – no (%)			
Symptomatic	4 (22.2%)	22 (19.8%)	0.8140
Asymptomatic	14 (77.8%)	89 (80.2%)	0.8140
Trimester of Maternal Zika infection – no. (%)			
T1 infection	7 (38.9%)	27 (24.3%)	0.1930
T2 infection	7 (38.9%)	37 (33.3%)	0.6450
T3 infection	2 (11.1%)	25 (22.5%)	0.2200
Unknown	2 (11.1%)	22 (19.8%)	0.3040
Dichorionic twins	2 (11.1%)	6 (5.4%)	0.3090
Fetus gender – no. (%)			
Female	10 (55.6%)	63 (56.8%)	0.9240
Male	8 (44.4%)	48 (43.2%)	0.9240
Term at birth (weeks' gestation) - median (min-max)	39 (32-41)	39 (36-41)	0.8661
Premature birth <37wg	1 (5.6%)	6 (5.4%)	0.6600
Mode of delivery – no. (%)			
Normal	13 (72.2%)	97 (87.4%)	0.0920
C-section	5 (27.8%)	14 (12.6%)	0.0920
Neonatal adaptation			
Abnormal Apgar score (<6 at 5 min) – no (%)	2 (11.1%)	13 (11.7%)	0.6510
Abnormal Lactate level (> 4.5) – no (%)	2 (11.1%)	16 (14.4%)	0.5230

[‡] including multiple maternal co-morbidities

* one primary cytomegalovirus infection in the infected group; one primary CMV and one primary toxoplasmosis

Alcohol consumption was defined as ongoing consumption during the pregnancy after its diagnosis

Current smoking was defined as ongoing smoking during pregnancy after its diagnosis

Trimester of maternal Zika infection was estimated based on symptoms onset, or on laboratory results (asymptomatic)

As the exposed group is small, we did not observe significant differences between the baseline characteristics. However, the approach using standardized differences permits us to highlight unbalanced characteristics, for which we tested and adjusted the risk estimates. Thus, we believe that keeping this approach for our study seems to be the most appropriate method of addressing potential confounding factors and effect-modifiers.

As proposed by the reviewer, Table 2 is now also presented as forest plot in a supplementary figure.

Supplementary figure with Relative risks and 95%CI for each main outcome:

The variables presented in tables 3 and 4 were not part of our primary outcomes, which is why we did not present association tests on these secondary outcomes (recommendations of the ASA, 2019). To fulfill the advice of the reviewers, p-values of Chi2, Fischer and t-tests have been added to Tables 3 and 4.

Table 3: Neonatal and early infantile outcomes, from birth to 2 months of life:

Neonatal and early infantile outcomes - From birth to 2 months of life	Confirmed congenital infection at birth N= 18	Negative testing at birth N= 111	p
Status at two months of live - no (%)			
Alive	17 (94.4%)	111 (100.0%)	0.1400
Neonatal demise	1 (5.6%)	0 (0.0%)	0.1400
Microcephaly < -3SD			
at birth* - no (%)	2 (11.1%)	1 (0.9%)	0.0077
at 2months** - no (%)	2/17 (11.8%)	1 (0.9%)	0.0077
Weight <-2SD			
at birth - no (%)*	2 (11.1%)	12 (10.8%)	0.9256
at 2months** - no (%)	2/17 (11.8%)	8/111 (7.2%)	0.5656
Structural brain anomalies - no (%)	6 (33.3%)	3 (2.7%)	<0.0001
Cortical development anomaly	4 (22.2%)	0 (0.0%)	<0.0001
Corpus callosum anomaly	4 (22.2%)	2 (1.8%)	0.0030
Calcifications or cystic lesions	5 (27.8%)	1 (0.9%)	<0.0001
Posterior fossa anomaly	4 (22.2%)	0 (0.0%)	<0.0001
Ventriculomegaly	4 (22.2%)	1 (0.9%)	0.0010
Ocular anomalies - no (%)			
Microphthalmia	1 (5.6%)	0 (0.0%)	0.1400
Fundoscopy anomalies	3/10 (30.0%)	2/34 (5.9%)	0.0330
Subretinal hemorrhage	2/10 (20.0%)	1/34 (2.9%)	0.0599
Chorioretinal lacunae	2/10 (20.0%)	1/34 (2.9%)	0.0599
Macula atrophy	1/10 (10.0%)	1/34 (2.9%)	0.3462
Abnormal otoacoustic emission - no (%)	2/12 (16.7%)	1/43 (2.3%)	0.1170
Severe neurologic symptoms - no (%)	5 (27.8)	1 (0.9%)	<0.0001
Arthrogryposis	1 (5.6%)	0 (0.0%)	0.1400
Hypertonia	3 (16.7%)	0 (0.0%)	0.0020
Dysphagia / swallowing disorders	2 (11.8%)	0 (0.0%)	0.0190
Seizures	1 (5.6%)	1 (0.9%)	0.2610
NICU Admission – no (%)	3 (16.7%)	14 (12.6%)	0.6372

¹ 128 alive infants evaluated at 1 and 2 months of life

* According to Intergrowth21 charts

**According to WHO Child Growth Standards

p -values were estimated by Chi2, Fischer or Wilcoxon tests

Table 4: Outcomes up to 3 years of life

Children outcomes - up to 3 years of life	Confirmed congenital infection at birth N= 15	Negative testing at birth N= 97	p
Microcephaly <-3SD*			
at 1 year	2 (13.3%)	1 (1.0%)	0.0060
at 2 years ¹	2 (13.3%)	1/96 (1.0%)	0.0080
at 3 years ²	2/11 (9.1%)	1/51 (2.0%)	0.0237
Weight < 5th percentile*			
at 1 years	2 (13.3%)	6 (6.2%)	0.2908
at 2 years ¹	2 (13.3%)	6/96 (6.3%)	0.2947
at 3 years ²	1/11 (9.1%)	3/51 (5.9%)	0.6944
Neurologic impairments at 2y ¹ – no (%)	5 (33.3%)	4/96 (4.2%)	0.0001
Cerebral palsy	2 (13.3%)	0/96 (0.0%)	0.0170
Severe dystonia or tremors	3 (20.0%)	3/96 (3.1%)	0.0070
Seizures	3 (20.0%)	2/96 (2.1%)	0.0170
Motor acquisitions			
Age at sitting position (m) - median (min-max)	6 (3-24)	6 (4-11)	0.8113
Delay for sitting position (>9m) – no (%)	2 (13.3%)	1 (1.0%)	0.0060
Age at walking (m) - median (min-max)	11 (8-24)	11 (7-17)	0.3289
Delay for walking ¹ (>18m) – no (%)	1 (6.7%)	0/96 (0.0%)	0.0110
Vision and hearing evaluation			
Impaired response to visual stimuli – no (%)	2 (13.3%)	1 (1.0%)	0.0060
Impaired response to auditory stimuli – no (%)	2 (13.3%)	1 (1.0%)	0.0060
Age at CDAS evaluation ² (m) - median (min-max)	35 (33-39)	36 (34-40)	0.7665
Global assessment – no (%)			
“Comfort” zone (>-1SD)	3/11 (27.3%)	30/51 (58.8%)	0.0572
“To be monitored” zone ([-2SD;-1SD])	1/11 (9.1%)	14/51 (27.5%)	0.1972
“Referral” zone (<-2SD)	7/11 (63.4%)	7/51 (13.7%)	0.0003
Motor domain – no (%)			
“Comfort” zone (>-1SD)	7/11 (63.4%)	47/51 (92.2%)	0.0105
“To be monitored” zone ([-2SD;-1SD])	2/11 (18.2%)	3/51 (5.8%)	0.1742
“Referral” zone (<-2SD)	2/11 (18.2%)	1/51 (2.0%)	0.0790
Socio-emotional domain – no (%)			
“Comfort” zone (>-1SD)	7/11 (63.4%)	41/51 (80.4%)	0.2280
“To be monitored” zone ([-2SD;-1SD])	0/11 (0.0%)	4/51 (7.8%)	0.3369
“Referral” zone (<-2SD)	4/11 (36.4%)	6/51 (11.8%)	0.0442
Cognitive and language domain – no (%)			
“Comfort” zone (>-1SD)	3/11 (27.3%)	32/51 (62.7%)	0.3140
“To be monitored” zone ([-2SD;-1SD])	2/11 (18.2%)	16/51 (31.4%)	0.3820
“Referral” zone (<-2SD)	6/11 (54.5%)	3/51 (5.9%)	<0.0001

¹ 111 infants evaluated at 2years of life, including 15 with laboratory confirmed congenital ZIKV infection

² 62 children evaluated at 3 years of life using the CDAS, including 11 with laboratory confirmed congenital ZIKV infe

* according to WHO Child Growth Standards <2 years and CDC growth charts > 2 years of life

p -values were estimated by Chi2, Fischer or Wilcoxon tests

With these unadjusted association tests, we observe higher proportions of structural brain abnormalities, microcephaly, neurological symptoms, neurologic impairments, delayed motor acquisition and suspected neurodevelopmental delay in children who tested positive at birth for congenital ZIKV infection. Overall, these association tests support our analysis of main outcomes, adjusted for maternal infection in the first trimester and controlled for potential effect-modifiers, but do not provide additional information.

If the reviewers and editors feel that it is necessary to present adjusted analyses for all these variables, we will provide them, although they do not provide additional information and are contrary to current ASA recommendations, as these variables are not all part of our main outcomes.

To simplify the reading of tables 3 and 4, and following reviewers and editors advices, we have removed the raw values for weight and head circumference from the tables and presented them in a box-and-whisker plot (now Figure 2):

4. There are many opportunities for editing the text and syntax.

Following this advice, the manuscript has been sent to a professional translator to improve the text and syntax.

Reviewer #2 (Remarks to the Author):

The manuscript by Heini and colleagues presents the results from a prospective cohort study of 129 infants born of zika-infected mothers in French Guiana. Evaluations are conducted at 4 timepoints: birth, 2 months, 2 years, 3 years.

The selection procedures for enrolment in the cohort and follow-up are clearly described and the flowchart (figure 1) helps understanding the sample size at the different timepoints. The statistical methods used are adapted and well presented. Statistical comparison of baseline characteristics, between zika-infected and zika-non-infected children at birth, should however be conducted and the tests chosen mentioned in the statistical analysis section. I would recommend chi-2 tests for the categorical variables and non-parametric Wilcoxon tests for continuous variables. Thus, p-values could be added in Table 1 and in the results section. Indeed, in the results section, the authors compared proportions between infected and non-infected children, but without p-values it is not intuitive to identify whether there are statistical differences or not.

We thank the reviewer for their very positive comment and for their help to improve our manuscript. As explained in the response to the first reviewer, we did not present association tests in the first table, as we preferred to compare baseline characteristics using standardized differences, which are more appropriate in this study with a small exposed group. This approach has been clarified in the methods and is presented in Table 1 (see our responses above). Standardized differences allowed us to highlight potential confounding factors and interactions that were not "significant" using Chi-2, Fisher and Wilcoxon tests. Our risk estimates for main outcomes were adjusted and tested for these unbalanced characteristics.

Because the number of children for the evaluations at 2 months, 2 years and 3 years is not constant, I would recommend to start each paragraph in the results section but stating the sample size. The authors attempted to do so, but it was not systematic.

We have modified the paragraphs of outcomes at two and three years of life to clarify the sample size:
- *“At 2 years of life, 15 children with a confirmed congenital ZIKV infection at birth and 96 who tested negative at birth were available for follow-up at the CHOG pediatric clinic. Among infected children, 5/15 (33.3%) had neurologic impairments: 2 with cerebral palsy, 3 with severe dystonia, and 3 with seizures. Two of the those with neurologic impairments had motor acquisition delays, partial or complete blindness, and one had hearing deficits. Hearing impairment was also diagnosed in another infected child without neurologic impairments (Table 4). Overall, the risk of adverse outcomes at two years of life was higher in infected children (6/15, 40.0%) compared to those infants that tested negative at birth (5/96, 5.2%), even when only considering children without structural brain anomalies (2/10, 20.0% vs 3/93, 3.2%): aRR 6.7 [95%CI 2.2-20.0] and aRR 6.2 [1.2-33.0], respectively (Table 2).”*
- *“Eleven (11/17, 64.7%) children of the infected group and 51 (51/111, 45.9%) children of the group that tested negative at birth came for neurodevelopment screening in August and September 2019”.*

In Table 2, it is unclear how the authors could estimate p-values in the sub-analysis in neonates with structural brain anomalies as no neonate adverse outcome was observed in non-infected children, and they do not report a relative risk. The same way for demise. If the authors performed a chi-2 test, this should be reported. The low number of children with structural brain anomalies makes it difficult to evidence statistical differences; and as a consequence, lack of statistical difference may only be due to the small sample size. I wonder if presenting this sub-analysis is relevant.

We thank the reviewer for their remark. If no neonate had an adverse outcome in a group, we estimated p-values with exact Chi-2 tests, which is now mentioned in the footnotes for Table 2: *“If an adverse outcome was not observed in one of the groups, the p-value was estimated using a Fischer test”*

In our analysis for potential interactions, brain anomalies appeared to be an effect modifier for the other main outcomes. Additionally, in a previous version of this manuscript, the reviewers asked us to present a sub-analysis accounting for brain anomalies, as it was a relevant effect-modifier in other cohorts (Brasil, Nature med, 2019; Mulkey, Jama ped, 2019).

We recognize that our confidence intervals are wide, but even with the small number of cases, this sub-analysis shows that infected children without brain abnormalities also have a significantly higher risk of adverse childhood outcomes compared to uninfected children. Thus, the potential clinical implications of this sub-analysis seem important enough to maintain it.

In Table 3 and 4, p-values would be helpful to better apprehend the differences between the infected and non-infected children. This would be helpful as results presented in tables 3 and 4 are hardly mentioned in the results section and are not really discussed.

As presented in the response to the first reviewer, we have added p-values in Tables 3 and 4 to fulfill your advice (see above).

The statistical analyses were adjusted for co-factors, but the effect of some cofactors would have been of interest to the readers, especially the timing of the infection during pregnancy in the mothers.

We acknowledge that the timing of maternal infection is of great interest for this analysis. Main outcomes in Table 2 were adjusted by maternal infection in the first trimester, but we did not present the effect of this covariate. To fulfill this comment, we have added the effect of maternal infection in the first trimester versus maternal infection in the second or third trimesters in a stratified analysis (supplementary Table 2):

Main outcomes	Confirmed congenital infections	Negative neonatal testing	RR [95%CI]	p
Neonatal and early infantile adverse outcomes¹	8/18 (44.4%)	4/111 (3.6%)	12.3 [4.1-36.8]	<0.001
- maternal infection in the 1st trimester	6/7 (85.7%)	2/27 (7.4%)	11.6 [2.9-45.2]	} 0.7200
- maternal infection in the 2nd or 3rd trimester	2/11 (18.2%)	2/84 (2.4%)	7.6 [1.2-48.9]	
Adverse outcomes at 2 years of life²	6/15 (40.0%)	5/96 (5.2%)	7.7 [2.7-22.1]	<0.001
- maternal infection in the 1st trimester	3/6 (50.0%)	2/20 (10.0%)	5.0 [1.1-23.3]	} 0.6239
- maternal infection in the 2nd or 3rd trimester	3/9 (33.3%)	3/76 (3.9%)	8.4 [2.0-35.8]	
Referral for suspicion of neurodevelopment <-2SD in at least one domain at 3 years of life³	7/11 (63.6%)	7/51 (13.7%)	4.6 [2.0-10.5]	<0.001
- maternal infection in the 1st trimester	3/4 (75.0%)	3/15 (20.0%)	3.8 [1.2-12.0]	} 0.7010
- maternal infection in the 2nd or 3rd trimester	4/7 (57.1%)	4/36 (11.1%)	5.1 [1.7-15.8]	

¹ 129 infants evaluated from birth to 2 months of life

² 111 children evaluated up to 2 years of life

³ 62 children evaluated at 3 years of life using the Child Development Assessment Scale

Stratified analysis presented according to Mantel-Haenszel methods, } p-value referred to the test of homogeneity between the two strata

In this stratified analysis, we observed a higher proportion of children with adverse outcomes at two months, two years and three years of life after maternal infection in the first trimester of pregnancy, although this difference was not significant given the small size of the groups. As our study does not appear to be sufficiently powered make conclusions from this analysis, we prefer to keep it as a supplementary table.

In the statistical analysis section, the authors mentioned that stratified analysis would be performed in case interaction was evidenced. However, in the results section, no reference is made to interactions. Were interactions tested but found to be not significant? If this is the case, I would recommend mentioning it.

We tested all unbalanced characteristics as potential effect-modifiers and we did not identify any interactions except for brain structural abnormalities, which is why this sub-analysis was presented. To clarify, we have modified the results to read:

“Effect-modifiers

Exposures during pregnancy, maternal age, co-morbidities, socio-economic status, infant sex, twins, prematurity and the mode of delivery were tested as effect-modifiers on main outcomes and no interactions were identified. The presence of structural brain anomalies, however, was an effect-modifier for severe neurological symptoms at two months of life and adverse outcomes at two and three years of life. A sub-analysis of children with and without structural brain anomalies is presented in Table 2.”

The conclusions are supported by the results presented, but I would have liked the effect of the timing of the infection during pregnancy in the mothers to be discussed.

To fulfill this suggestion, we have modified the discussion to read:

“The impact of the trimester of maternal infection is contradictory in some studies. In the cohort from Rio de Janeiro, the authors found that adverse outcomes were not correlated with the trimester of maternal infection.³ Other cohorts have identified higher rates of brain structural anomalies and congenital Zika syndrome in cases of maternal infection in the first trimester^{16,20}. In our study, we observed a higher proportion of infants with neonatal, early infantile, or adverse outcomes at two or three years of life after maternal infection in the first trimester of pregnancy, although this difference was not significant as our study does not appear to be sufficiently powered to conclude on this covariate (Supplementary Table 2).”

The discussion about lost to follow-up patients is interesting. However, the authors state that the proportions are similar in both groups, but no p-values is provided to convince the readers that this is true. And it feels like the proportion of lost to follow-up is higher in non-infected children. Again, giving p-values would convince that the samples are not biased due to a higher proportion of lost to follow-up in a group.

Between two months and three years of life, six (6/17, 35.3%) and sixty (60/111, 54.1%) infants were lost to follow-up in the infected and non-infected groups, respectively (Figure 1). Using a Chi2 test, this difference is not significant: $p=0.1494$. The lack of significance may be due to the small number of patients infected. Using a standardized difference, we observe a difference of 0.38, which is considered “unbalanced”. The proportion of those lost to follow-up in the non-infected group could be higher as parents of asymptomatic children may have fewer clinical concerns. This hypothesis would have overestimated the absolute risk of adverse infantile outcomes in the non-infected group, resulting in an underestimation of the relative risks associated with congenital infection confirmed at birth on childhood adverse outcomes.

To fulfill this comment, we have modified the discussion to read:

“In our study, although not significant, the proportion of loss to follow up was higher among children who tested negative at birth compared to those with a confirmed congenital infection (60/111, 54.1% vs 6/17, 35.3%, $p=0.1494$, Std diff= 0.38), which suggests a potential selection bias on the outcome. Yet, it is difficult to know if the loss to follow up has selected the more severe cases or not. One would argue that the lack of clinical concern by parents, particularly in asymptomatic cases, might have driven the loss to follow-up. This would have overestimated the absolute risks of infantile adverse outcomes and the suspicion of neurodevelopment delay in the cohort, particularly in those that tested negative at birth. Thus, absolute risk in this study should be considered carefully.”

In conclusion, the authors provide valuable data, analysed using adequate statistical methods,

on the evolution of children exposed to zika virus during pregnancy and compare the evolution based on their infection status at birth.

We thank the reviewer for their very positive conclusion and we hope that we have responded adequately to all their remarks.

Editorial requests:

- **Editorial policy checklist:** completed
- **Reporting summary:** completed
- **Custom software/code:** completed. This study does not include custom software. Codes (.log) have been provided in a zip file.
- **Data and code availability:** included in the first version of the manuscript:
 - o *“Source data that underlie the results are provided with this paper. Other individual participant data and codes will be shared with researchers who provide a methodologically sound proposal for multi-centric study, particularly individual participant data meta-analysis. Proposals should be directed to leo.pomar@chuv.ch.”*
- **Please replace your bar graphs with plots that feature information about the distribution of the underlying data. All data points should be shown for plots with a sample size less than 10. For larger sample sizes, please consider box-and-whisker or violin plots as alternatives. Measures of centrality, dispersion and/or error bars should be plotted and described in the figure legend:**
 - The Figure 2, presenting head circumferences and weights, is now a box-and-whisker graph.
 - The Figure 3 is a bar graph presenting the raw numbers and percentages of children evaluated through the Child Development Assessment Scale. As this figure is only descriptive and does not present measures of centrality or dispersion, we believe that a bar graph is appropriate. However, if the editors think that another graph may be more appropriate, we would be pleased to make any additional changes.

REVIEWERS' COMMENTS

Reviewer #1 (Remarks to the Author):

I wish to thank the authors for their ardent and thorough responses. I do think that ultimately they have done the correct analysis, but perhaps missed the point that both reviewers were making. We concur with the ASA and have not asked them to perform otherwise. Precisely our point: analyze in univariate, control for significance of univariate in multivariate.

They have now more or less done so. I might introduce a key point of caution in response to their timing of maternal infection. Remote diagnosis of primary maternal ZIKV infection, in the absence of good IgG isotype testing, is nearly impossible. I would caution against assumptions of seroconversion with respect to maternal trimester of pregnancy, since they simply cannot know with certainty.

The take home message is clear: if either the mom (in any trimester) or the placenta tests positive, then there is higher risk of long term adverse outcomes. Assure this message comes through as it is of crucial public health importance and advocates for ongoing maternal testing.

I am happy as it stands now.

Reviewer #2 (Remarks to the Author):

The authors have addressed my previous queries adequately.

I have no other queries.

Reviewer #1 (Remarks to the Author):

I wish to thank the authors for their ardent and thorough responses. I do think that ultimately they have done the correct analysis, but perhaps missed the point that both reviewers were making. We concur with the ASA and have not asked them to perform otherwise. Precisely our point: analyze in univariate, control for significance of univariate in multivariate.

We thank the reviewer for this positive comment. Indeed, whether using association tests or standardised differences, we agree that it is important to control for potential confounding factors,

which we have strengthened in our revised version. We have illustrated with a concrete example that the use of standardised differences seems to be preferred for our study, and we have indeed controlled unbalanced baseline characteristics in a multivariate analysis.

I might introduce a key point of caution in response to their timing of maternal infection. Remote diagnosis of primary maternal ZIKV infection, in the absence of good IgG isotype testing, is nearly impossible. I would caution against assumptions of seroconversion with respect to maternal trimester of pregnancy, since they simply cannot know with certainty.

We recognise that it is difficult to accurately date maternal infection on the basis of serology without IgG isotype testing. To follow this recommendation, we believe it is clearer to define “the trimester of maternal infection diagnosis” rather than the trimester of maternal infection.

We have modified throughout the manuscript to read:

- Results, baseline characteristics: “*Median maternal age at delivery was 25 and 26 years-old in the group of congenital infections and the negative group, respectively. Maternal infection diagnosed in the 1st trimester of pregnancy was more frequent in mothers of infected newborns (38.9% vs 24.3%).*”
- Results, neonatal and early infantile outcomes: “*The risk of adverse outcomes at two months was higher for infected infants compared to those tested negative at birth (4/111, 3.6%), even after adjustment for maternal infection diagnosed in the first trimester of pregnancy: : aRR 10.1 [95%CI 3.5-29.0]*”
- Discussion, interpretations: “*In our study, we observed a higher proportion of infants with neonatal, early infantile or adverse outcomes at two and three years of life after maternal infection diagnosed in the first trimester of pregnancy, although this difference was not significant as our study does not appear to be sufficiently powered to conclude on this covariate (Supplementary Table 2). Moreover, our study reports the trimester at infection diagnosis but does not permit to accurately date maternal infection, as the diagnosis is based on serology in many cases.*”
- Methods, statistical analysis: “*Timing of maternal infection diagnosis was estimated based on symptom onset or on laboratory results in cases of asymptomatic infection; and grouped into 1st or 2nd and 3rd trimesters for the analysis.*” “*The Relative Risks (RR) associated with laboratory confirmed congenital ZIKV infection were assessed using generalized linear models, and were adjusted (aRR) for confounding factors (trimester at maternal ZIKV infection diagnosis), and controlled for potential interactions with exposures during pregnancy, maternal age, co-morbidities and socio-economic status, infant gender, twins, prematurity and the mode of delivery.*”

The take home message is clear: if either the mom (in any trimester) or the placenta tests positive, then there is higher risk of long term adverse outcomes. Assure this message comes through as it is of crucial public health importance and advocates for ongoing maternal testing.

We have modified the conclusion to fulfill this comment: “*Overall, the results from our study along with those from previously published studies seem to indicate that a laboratory confirmed congenital ZIKV infection at birth could be associated with higher risks of long term outcomes, even in children without structural brain anomalies. As a normal antenatal and neonatal evaluation cannot provide complete reassurance for children exposed to ZIKV in utero, it seems paramount to offer systematic testing for*

congenital ZIKV infection at birth in cases of in-utero exposure, and to adapt counseling according to these results.”

I am happy as it stands now.

We thank the reviewer for their careful comments, which have improved this manuscript considerably.

Reviewer #2 (Remarks to the Author):

The authors have addressed my previous queries adequately.

I have no other queries.

We thank the reviewer for this positive comment and for their help to improve our manuscript.